# Novel Sophoridine Derivatives as Potential Larvicidal Agents against *Aedes albopictus*: Synthesis, Biological Evaluation, Acetylcholinesterase Inhibition, and Morphological Study

**DOI:** 10.3390/insects14040399

**Published:** 2023-04-20

**Authors:** Song Ang, Nana Cao, Wende Zheng, Zhen Zhang, Jinxuan Li, Zhenping Yan, Kaize Su, Wing-Leung Wong, Kun Zhang, Weiqian David Hong, Panpan Wu

**Affiliations:** 1School of Biotechnology and Health Sciences, Wuyi University, Jiangmen 529020, China15875045599@163.com (W.Z.); kzhang@gdut.edu.cn (K.Z.); 2International Healthcare Innovation Institute (Jiangmen), Jiangmen 529040, China; 3The State Key Laboratory of Chemical Biology and Drug Discovery, Department of Applied Biology and Chemical Technology, The Hong Kong Polytechnic University, Hung Hom, Kowloon, Hong Kong SAR, China

**Keywords:** sophoridine derivatives, synthesis, *Aedes albopictus*, biological evaluation, structure–activity relationship, acetylcholinesterase inhibition, morphological study

## Abstract

**Simple Summary:**

Sophoridine, a natural quinolizidine alkaloid with biological activities, was isolated and identified from traditional medicinal herbs. Although mosquitoes can be effectively controlled by chemical insecticides, their intense use has significantly increased the development and spread of resistant mosquitoes. Chemical insecticides have negative effects, such as potential health risks, water contamination, environmental pollution, and toxicity to nontarget organisms, including people. In this study, two series of sophoridine derivatives were generated using sophoridine as the lead compound. All derivatives were characterized and screened for their insecticidal activity against larva and adult mosquitoes. Their effects on the growth cycle of larva and their action on AChE from *Aedes albopictus* were also explored. Finally, morphological studies of the derivative-treated dead larvae were conducted.

**Abstract:**

Two series of novel sophoridine derivatives were designed, synthesized, and evaluated for their anti-mosquito activity. **SOP-2g**, **SOP-2q**, and **SOP-2r** exhibited potential larvicidal activity against *Aedes albopictus* larva with LC_50_ values of 330.98, 430.53, and 411.09 ppm, respectively. Analysis of structure–activity relationships indicated that the oxime ester group was beneficial for improving the larvicidal biological activity, whereas the long-chain aliphatic group and fused-ring group were introduced. Furthermore, the larvicidal mechanism was also investigated based on the inhibition assay of acetylcholinesterase (AChE) and the morphological observation of dead larva treated with derivatives. Results indicated that the AChE inhibitory activity of the preferred three derivatives were 63.16%, 46.67%, and 35.11%, respectively, at 250 ppm concentration. Additionally, morphological evidence demonstrated that **SOP-2q** and **SOP-2r** induced changes in the larva’s intestinal cavity, caudal gill, and tail, thereby displaying larvicidal action against *Ae. albopictus* together with AChE inhibition. Therefore, this study implied that sophoridine and its novel derivatives could be used to control the population of mosquito larva, which may also be effective alkaloids to reduce the mosquito population density.

## 1. Introduction

Mosquitoes such as *Aedes albopictus*, *Culex quinquefasciatus*, and *Anopheles gambiae* act as vectors that transmit pathogens and cause parasitic and viral infectious diseases (e.g., malaria, dengue, yellow fever, Japanese encephalitis, Chikungunya, West Nile virus, lymphatic filariasis, and Zika viruses), posing an ongoing threat all over the world. Tropical and subtropical regions are particularly vulnerable because the warm and humid climate creates ideal conditions for mosquito breeding [1,2,3,4,5]. For example, 120 million individuals have lymphatic filariasis transmitted by *Culex spiculosus*, and about 80% of them reside in tropical nations including Bangladesh, India, Indonesia, Malaysia, Nigeria, and the Philippines [6]. According to the World Health Organization, the parasitic infection of malaria transmitted by Anopheline mosquitoes causes an estimated 219 million cases worldwide and gives rise to more than 400,000 deaths annually. Unfortunately, children under 5 years old are the most vulnerable among them to suffer death [7,8]. Hence, mosquito control has become an urgent task in light of the present situation.

At present, utilizing larvicides and adulticides is one of the main techniques to reduce the mosquito population, whereas chemical insecticides are generally accepted as the most effective way to control mosquitoes [9,10,11,12,13]. The insecticides most commonly used to control vectors work by blocking voltage-gated ion channels (pyrethroids and organochlorines) or by preventing the vital enzyme acetylcholinesterase (AChE) from functioning (organophosphates and carbamates). It is known that AchE is a nervous system enzyme and is primarily responsible for the quick hydrolysis of acetylcholine (ACh) in the synaptic cleft, thereby controlling the timing of neurotransmission. ACh accumulates in the cholinergic binding site when AchE is inhibited and, consequently, causes persistent stimulation of nerve fibers through the peripheral and central nervous system, leading to the paralysis and death of the Ae. albopictus larvae [14,15,16,17]. Recently, Knutsson et al. [15] found that a number of noncovalent inhibitors target *An. gambiae* and *Ae. aegypti* mosquitoes’ AChE1 with specificity. However, the development and spread of resistant mosquitoes caused by the intense use of insecticides, as well as the negative effects of chemical insecticides (e.g., potential health risks, water contamination, environmental pollution, and toxicity to nontarget organisms, including people), have significantly increased [18,19,20]. Therefore, researchers are inspired to identify economical, effective, and target-specific strategies for isolating phytochemicals from plants with larvicidal properties or modifying the structures of natural products.

Natural phytochemicals offer an abundant source for controlling mosquito populations, such as pyrethrins, tannine, tarpeans, isofavinoides, and d-limonene [21,22,23]. These natural products are useful starting points for the development of effective anti-mosquito compounds [24]. Sophoridine, a natural quinolizidine alkaloid with biological properties such as anticancer, anti-inflammatory, and antivirus activities has been isolated and identified from traditional medicinal herbs, including *Sophora flavescens* Alt., *Sophora alopecuroides* L., and *Sophora viciifolia* Hance [25,26]. Sophoridine reportedly possesses insecticidal activity against lice, nematodes, aphids, and termites, and the research on the effect of sophoridine and its derivatives on insects has attracted the attention of scientists in recent years. The previous research has showed that the total alkaloid of sophora, including sophoridine, could kill 95% of the larvae [27,28,29,30]. However, few studies have been conducted on the anti-mosquito effect of sophoridine and its derivatives. A variety of biological activities and the good insecticidal activity of sophoridine and its derivatives make them potential candidates as mosquitocidal agents. Therefore, in the present study, we focused on the structural modification on the C14 position of sophoridine to generate a series of its derivatives. All derivatives were characterized and screened for insecticidal activity against larva and adult mosquitoes and explore their structure–activity relationship (SAR). Furthermore, their effects on the growth cycle of larva and their action on AChE from *Ae. albopictus* were explored to clarify the mechanism of action. Finally, morphological studies of derivative-treated dead larvae were conducted.

## 2. Materials and Methods

### 2.1. Instruments and Reagents

Sophoridine was purchased from Chengdu Durst Biotechnology Co., Ltd. Reagents were commercially available and used as received. All reactions were monitored by thin-layer chromatography (TLC; Qingdao Haiyang Chemical, Qingdao, China), and spots were observed with ultraviolet light. Column chromatography was performed on silica gel (200–300 or 300–400 mesh). NMR spectra were obtained on a Bruker DPX-500 MHz instrument. High-resolution mass spectra (HR-MS) were obtained on a Bruker micro-TOF-Q in ESI mode (Brooke, Switzerland).

### 2.2. Insects

Mosquito (*Ae. albopictus*, susceptible strain) colonies were collected at Huangpu District, Guangzhou, Guangdong Province, China, and had been maintained consecutively since 2013 in the laboratory. The mosquitoes were maintained in the laboratory of the International Healthcare Innovation Institute (Jiangmen), Jiangmen, China. Larvae were fed daily with fish food, and adults were placed in a rearing cage (30 × 30 × 30 cm) and received a 5% glucose solution. The mosquitoes were reared under 12:12 light/dark photoperiod and 70 ± 5% relative humidity at 26 ± 2 °C. The fourth-instar larvae female mosquitoes were used for the bioassay.

### 2.3. General Procedure for the Synthesis of SOP-1

To generate anhydrous and oxygen-free conditions, a magnetron was first introduced into a double-necked bottle. Nitrogen gas (N_2_) was then substituted for the air in the bottle. The double-necked flask was filled with 1.2 mL of anhydrous dimethyl formamide (DMF) and 5 mL of anhydrous CH_2_Cl_2_ (DCM) to serve as the solvent. Afterwards, POCl_3_ (2.30 g, 15.0 mmol) was slowly added to the reaction system with a syringe, and the reaction system was stirred at 0 °C for 1 h. Subsequently, 15 mL of anhydrous DCM solution was mixed with sophoridine (1.24 g, 5.0 mmol) and added dropwise to the reaction system using a syringe. The reaction system was then stirred at 0 °C with a gradual increase to room temperature. TLC was used to monitor the reaction. After roughly 9 h, the reaction was completed and the mixture was concentrated in vacuo to remove excess POCl_3_ and a large amount of DCM solution. After adding 5 mL of ice water, the pH of the reaction system was adjusted to 8–9 with 40% NaOH solution. Subsequently, the reaction system was hydrolyzed at 60 °C for 2 h and the reaction was monitored by TLC. After cooling the solution, the solvent was removed and extracted twice with ethyl acetate. The organic phases were combined, dried over anhydrous MgSO_4_, filtered, and concentrated under reduced pressure to obtain the crude product. Finally, the residue was purified by silica-gel column chromatography (*V*_MeOH_/*V*_DCM_ = 1/8) to obtain the compound **SOP-1** as a pale-yellow oil with 92% yield [31,32,33].

*Data for* **SOP-1:** ^1^H NMR (500 MHz, CDCl_3_) *δ* 9.64 (s, 1H), 3.63 (dd, *J* = 12.7, 5.6 Hz, 1H), 3.28 (td, *J* = 7.9, 4.0 Hz, 1H), 3.13–3.05 (m, 1H), 2.90–2.84 (m, 1H), 2.78 (dt, *J* = 11.8, 6.8 Hz, 1H), 2.37 (ddd, *J* = 15.9, 8.9, 3.5 Hz, 2H), 2.31–2.24 (m, 2H), 2.22 (dd, *J* = 11.1, 8.0 Hz, 1H), 2.12–2.02 (m, 1H), 1.84 (tdd, *J* = 13.3, 9.6, 5.9 Hz, 4H), 1.71 (qt, *J* = 12.4, 3.7 Hz, 1H), 1.60 (ddd, *J* = 13.4, 8.4, 3.5 Hz, 1H), 1.50 (qd, *J* = 11.0, 5.2 Hz, 4H), 1.05 (qd, *J* = 12.9, 4.2 Hz, 1H). ^13^C NMR (125 MHz, CDCl_3_) *δ* 187.38, 152.71, 110.02, 61.68, 60.41, 55.31, 52.21, 48.78, 39.78, 30.07, 24.81, 22.88, 22.76, 22.11, 19.85. HRMS (ESI): C_16_H_24_ClN_2_O (295.1572) [M+H]^+^ = 295.1568.

### 2.4. General Procedure for the Synthesis of SOP-2

Methanol (5 mL) and hydroxylamine hydrochloride (5.4 mmol) were added to a solution of NaOH (91.9 of mg, 2.3 mmol) in 10 mL of water. After the solution was evenly mixed, **SOP-1** (158.8 mg, 0.5 mmol) was slowly added and stirred at 0 °C. The reaction temperature was gradually returned to room temperature under TLC monitoring. After stirring for about 9 h, the reaction mixture was treated with water and extracted with ethyl acetate three times, dried with anhydrous MgSO_4_, filtered, and concentrated under reduced pressure to obtain the crude product. The residue was purified by silica-gel column chromatography (*V*_MeOH_/*V*_DCM_ = 1/8) to give a pale-yellow solid compound, **SOP-2**, with 90% yield [31,32,33].

*Data for* **SOP-2:** ^1^H NMR (500 MHz, CDCl_3_) *δ* 10.90 (s, 1H), 8.20 (s, 1H), 3.52 (dd, *J* = 11.6, 3.5 Hz, 1H), 3.01–2.91 (m, 4H), 2.83 (dd, *J* = 10.9, 4.8 Hz, 1H), 2.52 (d, *J* = 11.3 Hz, 1H), 2.40 (dd, *J* = 17.2, 6.3 Hz, 1H), 2.31–2.19 (m, 2H), 2.16 (t, *J* = 11.5 Hz, 1H), 2.11–2.05 (m, 1H), 1.96–1.68 (m, 6H), 1.53–1.43 (m, 2H), 1.33 (d, *J* = 13.0 Hz, 1H), 1.09 (qd, *J* = 12.9, 4.1 Hz, 1H). ^13^C NMR (125 MHz, CDCl_3_) *δ* 149.67, 142.79, 113.56, 63.77, 59.06, 54.08, 51.11, 45.12, 39.39, 29.29, 25.92, 25.85, 25.40, 25.27, 22.19, 18.62. HRMS (ESI): C_16_H_25_ClN_3_O (310.1681) [M+H]^+^ = 310.1681.

### 2.5. General Procedure for the Synthesis of SOP-2a-SOP-2r

A 200 mL round-bottom flask was charged with HOBT (212.2 mg, 1.0 mmol) and EDCI (191.7 mg, 1.0 mmol), and then 15 mL of DCM was added, dissolved completely, and stirred at room temperature for 3 min. Afterwards, **SOP-2** (150.0 mg, 0.5 mmol) was added and stirred for 3 h at room temperature. The reaction was monitored by TLC. The reaction mixture was poured into the water slowly (100 mL) and then extracted with ethyl acetate (3 × 40 mL). The organic phases were combined and dried with anhydrous MgSO_4_, filtered, and concentrated under reduced pressure to obtain the crude product. The residue was purified by silica-gel column chromatography (eluent: *V*_MeOH_/*V*_DCM_) = 1/8) to obtain **SOP-2a**-**SOP-2r** [31,32,33]. The compounds were produced in 68–94% yields, and their structures were determined with the aid of ^1^H and ^13^C NMR and high-resolution electrospray-ionization mass spectrometry (HRESIMS) data. Data for **SOP-2a** and **SOP-2b** are presented here, whereas those for **SOP-2c**-**SOP-2r** are in the Appendix A.

*Data for* **SOP-2a:** The compound was obtained in 80% yield as a white solid. ^1^H NMR (500 MHz, CDCl_3_) *δ* 8.60 (s, 1H), 8.09 (dd, *J* = 8.3, 1.2 Hz, 2H), 7.61–7.54 (m, 1H), 7.49–7.43 (m, 2H), 3.61 (dd, *J* = 12.2, 4.5 Hz, 1H), 3.06 (d, *J* = 11.6 Hz, 1H), 3.01–2.85 (m, 3H), 2.71 (dd, *J* = 11.2, 5.4 Hz, 1H), 2.65–2.59 (m, 1H), 2.52–2.43 (m, 2H), 2.37 (t, *J* = 11.9 Hz, 1H), 2.22 (dtt, *J* = 15.7, 7.4, 4.2 Hz, 1H), 1.98–1.92 (m, 1H), 1.91–1.82 (m, 2H), 1.79 (ddd, *J* = 13.7, 6.7, 3.8 Hz, 2H), 1.72 (dd, *J* = 12.8, 3.7 Hz, 1H), 1.66 (ddd, *J* = 16.8, 7.9, 3.5 Hz, 2H), 1.50 (dd, *J* = 12.6, 3.2 Hz, 1H), 1.34 (d, *J* = 12.6 Hz, 1H), 1.10 (qd, *J* = 12.7, 4.1 Hz, 1H). ^13^C NMR (125 MHz, CDCl_3_) *δ* 164.44, 156.81, 149.81, 146.92, 133.22, 129.71, 129.15, 128.56, 109.60, 106.67, 63.44, 59.63, 54.82, 51.71, 45.94, 40.17, 29.77, 26.19, 25.61, 25.51, 23.15, 19.28. HRMS (ESI): C_23_H_29_ClN_3_O_2_ (414.1943) [M+H]^+^ = 414.1947.

*Data for* **SOP-2b:** The compound was obtained in 72% yield as a white solid. ^1^H NMR (500 MHz, CDCl_3_) *δ* 7.89 (d, *J* = 8.0 Hz, 1H), 7.40 (td, *J* = 7.7, 1.3 Hz, 1H), 7.25 (t, *J* = 7.8 Hz, 2H), 3.59 (dd, *J* = 12.2, 4.5 Hz, 1H), 3.05 (d, *J* = 11.6 Hz, 1H), 2.99–2.94 (m, 1H), 2.95–2.81 (m, 2H), 2.68 (dd, *J* = 11.2, 5.4 Hz, 1H), 2.61 (s, 4H), 2.52–2.43 (m, 2H), 2.36 (t, *J* = 11.8 Hz, 1H), 2.20 (dq, *J* = 8.1, 5.7, 4.1 Hz, 1H), 1.95–1.81 (m, 3H), 1.78 (ddt, *J* = 14.6, 7.6, 3.7 Hz, 3H), 1.73–1.56 (m, 3H), 1.52–1.45 (m, 1H), 1.31 (d, *J* = 14.0 Hz, 1H), 1.09 (qd, *J* = 12.8, 4.2 Hz, 1H). ^13^C NMR (125 MHz, CDCl_3_) *δ* 165.47, 156.65, 146.86, 140.35, 132.15, 131.77, 130.25, 128.66, 125.78, 109.78, 63.46, 59.60, 54.77, 51.66, 45.90, 40.11, 29.71, 26.17, 25.59, 25.47, 24.75, 23.10, 21.57, 19.23. HRMS (ESI): C_24_H_31_ClN_3_O_2_ (428.2099) [M+H]^+^ = 428.2105.

### 2.6. General Procedure for the Synthesis of SOP-3

K_2_CO_3_ (107.0 mg, 0.8 mmol) was added to a solution of 200 mg **SOP-2** dissolved completely with 5 mL of DCM in a round-bottomed flask. The solution was stirred for 1 min. Then, m-CPBA (134.0 mg, 0.8 mmol) was added and the mixture was set in an ice bath. The reaction system was gradually returned to room temperature and stirred. TLC was applied to monitor the reaction. After 10 h, the mixture was suction filtered to remove excess K_2_CO_3_ and m-CPBA to obtain a crude product. This product was purified by silica-gel column chromatography (eluent: *V*_MeOH_/*V*_DCM_ = 1/8) to obtain a white solid (**SOP-3**) with 90% yield [34], as shown in Figure 1.

### 2.7. General Procedure for the Synthesis of SOP-3a-SOP-3g

HOBT (390.5 mg, 1.8 mmol), EDCI (352.7 mg, 1.8 mmol), pelargonic acid (142.4 mg, 0.9 mmol), and 15 mL of DCM were added into a 200 mL round-bottom flask, and the mixture was stirred at room temperature for 3 min. Then, **SOP-3** (292.6 mg, 0.9 mmol) was added into the reaction vessel under stirring at room temperature and monitoring by TLC. After reacting for 3 h, the mixture was treated with water and extracted with ethyl acetate three times. The organic phase was combined, dried, and filtered. The filtrate was concentrated under reduced pressure to obtain the crude product and purified by silica-gel column chromatography (eluent: *V*_MeOH_/*V*_DCM_ = 1/30) to obtain **SOP-3a**-**SOP-3f** [31,32,33]. These compounds were synthesized in 54–85% yields, and their structures were established by a combination of ^1^H and ^13^C NMR and HRESIMS data. Data for **SOP-3a** and **SOP-3b** are presented here, whereas those for **SOP-3c**-**SOP-3f** are in the Appendix A.

*Data for* **SOP-3a:** The compound was obtained in 54% yield as a yellow solid. ^1^H NMR (500 MHz, CDCl_3_) *δ* 8.51 (s, 1H), 8.01 (d, *J* = 6.4 Hz, 2H), 7.55–7.48 (m, 1H), 7.39 (t, *J* = 6.8 Hz, 2H), 5.30–5.10 (m, 1H), 3.66 (d, *J* = 8.4 Hz, 1H), 3.51 (t, *J* = 12.3 Hz, 2H), 3.37 (dt, *J* = 21.7, 5.4 Hz, 2H), 3.07 (dd, *J* = 16.8, 11.4 Hz, 2H), 2.89 (dd, *J* = 39.8, 9.5 Hz, 2H), 2.58–2.37 (m, 3H), 2.27 (q, *J* = 18.0, 15.2 Hz, 2H), 1.87–1.42 (m, 5H), 1.31–1.10 (m, 2H). ^13^C NMR (125 MHz, CDCl_3_) *δ* 164.23, 156.26, 146.12, 133.39, 128.61, 127.92, 113.07, 73.38, 70.27, 63.95, 56.95, 53.57, 50.56, 33.92, 31.74, 26.79, 25.28, 23.67, 22.12, 21.77, 19.85.

*Data for* **SOP-3b:** The compound was obtained in 56% yield as a yellow solid. ^1^H NMR (500 MHz, CDCl_3_) *δ* 8.48 (s, 1H), 7.79 (d, *J* = 16.0 Hz, 1H), 7.53 (dd, *J* = 6.4, 3.0 Hz, 2H), 7.44–7.30 (m, 3H), 6.49 (d, *J* = 16.0 Hz, 1H), 3.71 (d, *J* = 8.3 Hz, 1H), 3.61–3.51 (m, 2H), 3.48–3.37 (m, 1H), 3.16–3.05 (m, 2H), 3.05–2.96 (m, 1H), 2.86 (d, *J* = 9.4 Hz, 1H), 2.57 (qd, *J* = 15.5, 13.6, 7.7 Hz, 2H), 2.44 (ddd, *J* = 17.6, 11.3, 8.0 Hz, 1H), 2.35–2.25 (m, 2H), 1.91–1.76 (m, 4H), 1.75–1.60 (m, 2H), 1.54 (d, *J* = 13.1 Hz, 1H), 1.33–1.15 (m, 2H). ^13^C NMR (125 MHz, CDCl_3_) *δ* 164.89, 155.80, 146.08, 134.34, 130.67, 129.02, 128.28, 115.67, 112.93, 73.87, 70.93, 64.04, 57.25, 50.60, 33.86, 31.75, 27.00, 25.25, 23.81, 22.15, 21.77, 19.86. HRMS (ESI): C_25_H_31_ClN_3_O_3_ (456.2048) [M+H]^+^ = 456.2041.

### 2.8. Bioassay

#### 2.8.1. Insecticidal Tests for Larvae of *Ae. albopictus*

According to the literature [35], each derivative was dissolved in acetone with a final concentration of 1000 ppm. An appropriate amount of live *Ae. albopictus* fourth-instar larvae (dark gray and about 7 mm long) were selected and washed two to three times with deionized water. A 24-well plate with a test well and four replication wells for each derivative was used, and each well had five larvae in it. Extra deionized water was removed with a pipette, and then 985 μL of clean deionized water and 5 μL of feed solution containing 25 mg/mL was added. Finally, about 10 μL of derivative solution was added. Deltamethrin (1000 ppm) and acetone replaced the derivative as positive and negative control groups, respectively. Three independent replicate tests were performed. The 24-well plate was cultivated in an incubator maintained at a constant temperature of 28 °C and 80% relative humidity under light and dark conditions for 12 h each. After 24 h, the lethality of each derivative to the larva was recorded if the larva did not move by using the pipette tip to touch the larva’s body, at which point it was assumed to be dead.

After the pre-experiment screening, derivatives with a lethality of at least 50% were chosen to participate in the semi-inhibitory concentration test. Each derivative was diluted with acetone to obtain five to nine concentration gradients ranging from 100 ppm to 1000 ppm. Subsequently, the above procedure was repeated, and the lethality of each derivative to the larva was recorded. According to the relationship between the concentration of sophoridine derivatives and mortality, the virulence regression equation was established. The LC_50_ value was calculated according to the equation below, and the larvae in the blank control group should not die off more than 5% given the variations in each individual:Mortality rate (%) = A1/B1 × 100%

A1: number of larvae killed at each concentration on the 24-well plate;

B1: initial number of larvae added per concentration on the 24-well plate.

#### 2.8.2. Insecticidal Tests for Female Mosquitoes of *Ae. albopictus*

In order to determine the insecticidal activity toward female mosquitoes of *Ae. albopictus*, a 0.25 μL (equivalent to 0.25 μg/female) of the 1000 ppm aforementioned derivative solution was topically dropped onto the dorsum of adult mosquitoes with a handheld pipetting device [36,37]. The derivatives were dissolved in acetone, a highly volatile substance that stayed on the epidermis of adult mosquitoes for a short period of time. Twenty-five susceptible female mosquitoes that were 2–5 days after eclosion and had not sucked blood were collected into each test cup. The female mosquito was lightly anesthetized with CO_2_ for 15–30 s and then placed on a cold table cooled to 4 °C in advance. A quick operation ensured that the mosquito remained anesthetized during the procedure. Three parallel groups were made for each test, and the female mosquitoes were treated with 0.25 μL of pure acetone as a negative control. After drug administration, the mosquitoes were transferred into another test cup with cotton moistened with a 10% sugar water solution and placed on the top of the test cup. Mortality was recorded after 24 h of rearing at 26–28 °C, 80% relative humidity, and light/dark (12 h/12 h). Deltamethrin (0.25 µg/female) and acetone were used as positive and negative control groups, respectively. Importantly, the mortality rate of the negative control group did not exceed 5%. Three sets of repeated tests were conducted for different batches of adult mosquitoes. Finally, the lethality rate of the derivatives for adult mosquitoes was calculated by the following formula:Lethality rate (%) = A2/B2 × 100%

A2: number of female adult mosquitoes killed in each test cup;

B2: number of female adult mosquitoes added into each test cup.

#### 2.8.3. Effects on the Growth Cycle of *Ae. albopictus* Fourth-Instar Larvae

##### Effects on the Emergence of *Ae. albopictus* Larvae

Compounds **SOP-2g**, **SOP-2q**, and **SOP-2r**, which had better larvicidal activity than other sophoridine derivatives, were screened out to investigate their effects on the emergence of *Ae. albopictus* larvae. The final test concentration of the derivative was set at LC_30_ according to the result of the LC_50_ test and using the high-throughput screening method [38,39]. A total of 5 fourth-instar larvae of *Ae. albopictus* with 985 μL of deionized water, 10 μL of sample solution, and 5 μL of feed solution were added to each well on a 24-well plate and incubated under constant temperature and humidity for 24 h. Eight replicate wells were set for each concentration, and three independent replicate experiments were performed. After culturing for 24 h, the still-surviving larvae were sucked out with a dropper and rinsed with deionized water two to three times. Afterwards, the larvae were transferred onto a new 24-well plate, with one treated larva placed in each well with 985 μL of deionized water, 10 μL of the same sample solution, and 5 μL of feed solution added. The 24-well plate of larvae treated with the same compound was laid into a mosquito cage, and 10% sugar water was handed into each cage concurrently. The condition of the rearing environment was set at 28 °C, the relative humidity was 80%, and it was light and dark for 12 h each. Finally, the growth state of the larvae was observed every 12 h from the day the larvae were placed into the mosquito cage. The status of the larvae was recorded with the following scores: 0, death; 4, larva; 5, pupa; 6, adult mosquito; 4-0, death as larva; 5-0, death as pupa; and 6-0, death as an adult mosquito.

##### Effects on the Fertility of Female Adult Mosquitoes of *Ae. albopictus*

The mosquitoes, evolving from the larvae that survived in the above experiments of effects on the emergence of larvae, were starved for 24 h. They were then fed with blood, and the abdomen of the female mosquitoes was clearly bulging and blood red after blood sucking [40]. At this time, the female mosquitoes were transferred into new cages by using a manual suction device. Five female mosquitoes, an egg-collection device, and a water-feeding device were placed inside each cage. The average number of eggs laid by females was used as an indicator to calculate the fertility of females with the formula below:Average number of eggs laid (%) = number of eggs on oviposition paper/number of females laying eggs × 100%

#### 2.8.4. Inhibitory Activity of Compounds against Larval Enzyme

According to the reported method [41], 50 fourth-instar larvae of *Ae. albopictus* to be tested were initially aspirated with a dropper. The larvae were washed twice with deionized water, and the water on the surface of the larvae was blotted with filter paper. Second, the larvae were transferred into a 1.5 mL centrifuge tube. Then, 0.6 mL of ice-cold 0.1 M PBS buffer (pH 8.0) was added to the tube and ground in an ice bath with a grinding rod for 30 s. Third, the larvae were sonicated 20 times at low temperature with an ultrasonic crusher. Finally, the tube was centrifuged at 1700× *g* for 15 min at 4 °C to remove the larval tissue fragments, and the supernatant was aspirated as the AchE solution to be tested. Then, AchE in the fourth-instar larvae of *Ae. albopictus* was determined based on the Ellman method. The sophoridine derivatives **SOP-2g**, **SOP-2q**, and **SOP-2r** were diluted into different concentration gradients in which 1 μL of each sample of concentration gradient was dropped onto a 96-well plate. The negative control group was 1 μL of acetone solution. Subsequently, 79 μL of AchE solution was pipetted onto a 96-well plate and shaken at room temperature for 30 s to mix well. After the mixture was incubated at 28 °C for 10 min, 10 μL of 5,5′-dithiobinitrobenzoic acid solution and 10 μL of acetylcholine iodosulfide solution were added rapidly under light-proof conditions. Then, the 96-well plate was immediately placed into the enzyme standard to determine the OD value at a 412 nm wavelength, and the OD value was measured every 1 min for 30 min. The activity was calculated by the following formula:Enzyme activity inhibition rate (%) = (1 − T/CK) × 100%
where T is the specific activity of enzyme in the treatment group and CK is the specific activity of the enzyme in the control group.
Specific activity of enzyme = enzyme activity unit/enzyme source protein content
Activity unit of enzyme = ∆OD_412_ × V/ε × L
where ∆OD_412_ is the change in absorbance per minute (∆OD_412_/min); V is the reaction system (μL); ε is the extinction coefficient [0.0136 L/μmoL/cm]; and L is the light range (1 cm).

#### 2.8.5. Observation of Morphological Changes in Dead Larvae

The morphology of instar larvae was observed according to a reported method [42]. First, the fourth-instar larvae of *Ae. albopictus* were treated with the LC_90_ concentration of sophoridine derivatives (**SOP-2q** and **SOP-2r**) for 24 h, whereas the control group was replaced with acetone. Second, three larvae were collected from each treatment group for morphological analysis and image acquisition. Third, the larvae were mounted onto glass slides and observed by an IX73 inverted fluorescence microscope (magnification, 4 × 10 and 8 × 10). Finally, the larval head, thorax, abdomen, and other organs such as tail gills, siphons, segments, antennae, eyes, and other organs were compared and observed, and images were collected.

## 3. Results and Discussion

### 3.1. Chemistry

A total of 27 derivatives, including three intermediates (**SOP-1**–**SOP-3**) and 24 target compounds, were prepared using commercially available sophoridine (**SOP**) as the starting material, as described in Figure 1. The target derivatives were divided into two series, which were obtained through a common two-step procedure. This procedure involved synthesizing the key intermediates of **SOP-1** by the Vilsmeier–Haack reaction of **SOP** with DMF in the presence of POCl_3_, and **SOP-2** was obtained by the reaction of compound **SOP-1** with NH_2_OH under the basic condition in MeOH/H_2_O. Then, **SOP-2a**–**SOP-2r** were synthesized by the reaction of compound **SOP-2** with different carboxylic acids (RCOOH) in good to excellent yields (68–94%). Additionally, N-oxide (**SOP-3**) was prepared by the oxidation of nitrogen atoms at position 1 in **SOP-2** by using mCPBA. Subsequently, the target compounds (**SOP-3a**–**SOP-3f**) were obtained using the same method by the reaction of compound **SOP-3** with different carboxylic acids (RCOOH) in 54–85% yields. All compounds were purified by silica-gel column chromatography, and TLC was used to monitor the experimental process. All products were characterized by FT-IR, ^1^H NMR, ^13^C NMR, and mass spectrometry.

### 3.2. Biological Evaluation

#### 3.2.1. Insecticidal Tests for Larval and Female Mosquitoes

Larvicidal activities against the fourth-instar larvae of *Ae. albopictus* of the synthesized compounds were investigated, and the structures of sophoridine and its derivatives with their mortalities are depicted in Table 1. The mortalities of the compounds at 1000 ppm ranged from 0% to 100%, which suggested that larvicidal activities may vary substantially with structural modifications. Oxime ester moiety is one of the most important fragments in organic chemistry and exists in lots of bioactive compounds that show a large number of bioactivities, such as antifungal, antioxidant, and insecticidal activities [43,44]. New matrine-type alkaloid analogs were designed by the introduction of oxime ester fragments into matrine at its C14 position and exhibited excellent agricultural activities against two threatening pests, *Tetranychus cinnabarinus* Boisduval and *Mythimna separata* Walker [45]. Therefore, this result suggested the importance of introducing the amidoxime ester fragment at the C14 position of sophoridine, because the mortalities of the derivatives were higher than those of the parent compound, especially compounds **SOP-2**, **SOP-2g**, **SOP-2q**, **SOP-2r**, **SOP-3**, and **SOP-3f**, with inhibition rates ranging from 80% to 100%. These values were much higher than the 10% inhibition rate of the parent compound. Additionally, the mortalities were lower than 50% when benzene ring structures, either with an electron-withdrawing group or an electron-donating group, were introduced into the oxime ester fragment. Meanwhile, when a long aliphatic alkyl chain with nine carbons or ten carbons was introduced, the inhibition rates increased to 85–100%, which were significantly higher than those of the parent compound. This finding indicated that α-naphthalene was an effective substitution to improve the larvicidal activity because the mortality reached 90% when the ligand was introduced into the oxime ester fragment.

Results of preliminary activity tests for all compounds against female *Ae. albopictus* mosquito are shown in Table 2. Sophoridine and its derivatives had low activity against adult mosquitoes, with the highest mortality of 35.27% less than 50% at test concentration. Therefore, the test of LC_50_ concentration for female mosquitoes of *Ae. albopictus* was not conducted in this study. The differences between larvae and female adult mosquitoes may be explained by the topical application of the compounds to evaluate insecticidal activity against adult mosquitoes. These compounds were required to infiltrate the epidermis and enter the adult mosquitoes, causing their disability or death. Conversely, larvicidal activity was tested by the microporous plate method, in which compounds were dissolved in water and entered the larva directly through feeding. However, the way of premedication could affect the results of insecticidal activity. For these series of SOP derivatives, their high polarity and hydrophilicity could lead to activity bias toward the larvae rather than the adult. Additionally, the bioassay conditions for larvae and adult mosquitoes are different. For example, whole larvae were immersed in 1000 ppm solutions; however, the female adult only topically accepted 0.25 µL 1000 ppm solution.

#### 3.2.2. Dose–Response Curves on *Ae. albopictus* Larvae

According to the result of preliminary activity tests on *Ae. albopictus* larvae, the dose–response curves of six selected sophoridine derivatives (**SOP-2**, **SOP-2g**, **SOP-2q**, **SOP-2r**, **SOP-3**, and **SOP-3f**) were determined with the expanded concentration test range of the compounds, as shown in Figure 1. Results indicated that the insecticidal efficacy on *Ae. albopictus* larvae showed a dose-dependent manner. As shown in Table 3, the LC_20_, LC_50_, and LC_90_ values on *Ae. albopictus* larvae were calculated according to the toxicity regression equations generated from the dose–response curves, and the R^2^ values ranged from 0.970 to 0.993, which were close to 1, indicating a good fit. Compounds **SOP-2g**, **SOP-2r**, and **SOP-2q** possessed high lethality (<500 ppm) to larvae, with LC_50_ values of 330.98, 411.09, and 430.53 ppm, respectively, whereas the LC_50_ of **SOP-3f** was 546.68 ppm. Similarly, the LC_50_ of the intermediate compound **SOP-2** was 527.01 ppm, which was lower than that of the intermediate compound **SOP-3** of 594.03 ppm. Overall, these results demonstrated that the N1 position of the parent compound sophoridine was an important active site.

### 3.3. Effects of Sophoridine on the Partial Life Cycle of Ae. albopictus

#### 3.3.1. Effects on the Emergence of *Ae. albopictus* Larvae

As shown in Figure 2, the effects of sophoridine derivatives (**SOP-2g**, **SOP-2q**, and **SOP-2r**) on larval emergence were tested and compared with the negative control group. The eclosion of the larvae in the negative control group started on the third day with a rate of 15%, whereas the larvae eclosion in the **SOP-2g** and **SOP-2r** treatment groups started on the fourth day with rates of 4% and 7%, respectively, which were five and three times less than that of the negative control group. The eclosion of the larvae treated with **SOP-2q** was further delayed and started only on the fifth day with an eclosion rate of 11%, whereas the eclosion rate of the negative control group reached 33% at this time point. The selected sophoridine derivatives delayed the emergence time and reduced the emergence rate of larvae of *Ae. albopictus*. Additionally, the mortalities of compound-treated groups increased in the first 3 and 5 days, in which the mortality rate of the **SOP-2g** treatment group was 7%, and those of the **SOP-2r** and **SOP-2q** treatment groups reached 13% and 15%, respectively. Meanwhile, no death occurred in the negative control group. Finally, the mortality rates of the test groups stabilized after 3 and 5 days in this experiment. These mortalities suggested that the chronic toxicity of the selected sophoridine derivatives caused the larvae to fail to transform into pupae and emerge successfully. Therefore, the result of the emergence experiment indicated the possibility of sustaining mosquito control by delaying the emergence time and reducing the emergence rate of larvae.

#### 3.3.2. Effects on the Fertility of Female Adult *Ae. albopictus* Mosquitoes

To explore the effect of SOP derivatives on the fecundity of *Ae. albopictus*, the average number of eggs laid by the adult mosquitoes that emerged from the SOP derivative-treated larvae was recorded. Results are shown in Figure 3, where it could be found that the average egg-laying rate of the treated female mosquitoes did not change significantly compared with that of the control group. Thus, the derivatives had no obvious effect on the fecundity of female *Ae. albopictus* mosquitoes.

### 3.4. SARs

The SAR analysis results of these novel sophoridine derivatives are summarized in Figure 4. First, the insecticidal activity against the larvae of *Ae. albopictus* significantly increased when the oxime ester fragment was introduced at the C14 position of sophoridine, whereas the parent compound showed low insecticidal activity. Second, a series of side-chain structures was introduced by esterification at the hydroxyl group of the oxime ester. Anti-mosquito activity did not improve when the side-chain groups were benzene-ring structures with either electron-absorbing or electron-donating groups compared with the parent compound. However, the larvicidal activity of the derivatives significantly increased when aliphatic long carbon chains and fused-ring fragments were introduced. Third, it was found that the N1 position of the parent compound sophoridine was an important active site to the larvicidal activity.

### 3.5. Inhibitory Activity of Compounds against AChE

The inhibition rates of the sophoridine derivatives (**SOP-2g**, **SOP-2q**, and **SOP-2r**) on AChE at different concentrations were tested, and results are shown in Figure 5. The inhibition rate of **SOP-2g** with 63.16% was higher than those of **SOP-2q** and **SOP-2r**, with 46.67% and 35.11%, respectively, at the highest concentration (250 ppm) of each compound. At the lowest concentration (50 ppm), **SOP-2g** remained the most potent analog with an inhibition rate of 44.72%. Intriguingly, the inhibitory activities against the larval AChE of these three sophoridine derivatives were generally concentration-dependent. Thus, the insecticidal mechanism of the SOP derivatives could be partially mediated by inhibiting AChE. Insect poisoning or even death can result from the cholinergic system being destroyed or obstructed, which overstimulates larval neurons and causes them to produce excess levels of the neurotransmitter acetylcholine. Further studies are required to validate this hypothesis.

### 3.6. Observation of Morphological Changes in Dead Larvae

#### 3.6.1. Observation of Larval Intestinal Cavity

The overall appearance of larvae in the negative control group was worm-like with eight obvious segments dividing the abdomen, as shown in Figure 6A. However, when the larvae were treated with the compounds, the abdomen segments became unclear, with multiple collapses laterally, and the abdomen became thinner and longer, as shown in Figure 6B,C. Moreover, the larvae of the negative control group had evenly distributed bristles, whereas the compound-treated groups had no continuous arrangement of bristles. According to the literature, the chemical toxicity of sophoridine derivatives may cause the deformation and hypertrophy of the intestinal epithelium of larvae, resulting in abdominal collapse. Then, the intestinal tract collapsed and noticeably darkened [46]. The first to fifth abdominal segments are in the midgut of the larva, playing a key role in the secretion of digestive enzymes [47]. Therefore, the imbalance in larval physiological function caused by the collapse of the intestinal lumen may explain the death of the larvae.

#### 3.6.2. Observation of Larval Caudal Gill

The caudal gills of larvae in the control group had a clear and cyst-like structure located at the end of the larvae, as shown in Figure 7A. Noticeably, Figure 7B,C show that the normal external structure existed with a shrunken internal structure after the larvae were treated with the sophoridine derivatives. The external cuticle of the caudal gill is reportedly extensively damaged when the internal structure of the larval caudal gill shrinks [48]. Additionally, the larval caudal gill participates in the uptake and elimination of most dysfunction-related ions to regulate larval electrolyte levels. The deletion and deformation of the caudal gill could lead to significantly reduced or lost larval uptake of sodium, potassium, chloride, and phosphate ions from the nutrient matrix. Therefore, the destruction of the caudal gill caused by the sophoridine derivatives may also be one of the reasons for the death of the larvae.

#### 3.6.3. Observation of Larval Tail

The tail of the larvae in the control group had a normal shape (Figure 8A), whereas the experimental treatment groups had intestinal content discharge at the tail (Figure 8B,C). According to the literature [47], the intestinal contents were expelled because the intestinal epithelial cells ruptured, thereby permitting the whole extrusion of the intestinal contents along with the nutritional matrix surrounding the intestine. Researchers have demonstrated that larvae underwent partial or complete elimination of nutritional substrates as a defensive strategy to remove hazardous chemicals [48,49]. The compound with a toxic effect on the larvae destroyed the intestinal epithelial cells, so the intestinal lumen completely collapsed. Moreover, the larvae started their own defense system that extruded the nutrient matrix out of the body, causing larval death.

These observations suggested that the intestinal cavity, caudal gill, and tail of the larvae were all subjected to varying degrees of injury as a result of the action of sophoridine derivatives, and there is a pattern to the morphological changes in the larvae following medication treatment. Other evidence points to the role of sophoridine derivatives in the disorganization of the larvae’s defense mechanism, the collapse of the intestinal lumen, and the loss of the caudal gill. These morphological alterations offer fresh avenues for further research into the mode and mechanism of sophoridine and its derivatives’ ability to harm larvae.

## 4. Conclusions

This study aimed to identify the lead compounds for anti-mosquito activity from structurally modified derivatives of the natural product sophoridine as alternatives to other synthetic insecticides currently on the market. Twenty-seven sophoridine derivatives were synthesized, and three derivatives (**SOP-2g**, **SOP-2q**, and **SOP-2r**) with good larvicidal activity were identified in anti-mosquito tests with LC_50_ values of 330.98, 430.53, and 411.09 ppm, respectively. Such activities were lower than commercially available decpermethrin and cypermethrin. However, it was first time studying the antimosquito activity of the natural product sophoridine against *Ae. albopictus*, and it is hoped that the negative effects of chemical pesticides, such as potential health risks, water pollution, environmental contamination, and toxicity to non-target organisms, will be addressed in further study. SAR relationship analysis of these novel sophoridine derivatives revealed that the introduction of the oxime ester fragment led to improved biological activity against *Ae. albopictus*. Aliphatic long carbon chains and fused-ring fragments were introduced at the hydroxyl group of the oxime ester by esterification, respectively, which improved the larvicidal activity. The larval emergence experiment of *Ae. albopictus* against these derivatives showed that larval emergence was delayed by 1–2 days, and the fledging rates decreased by 7–15%. The insecticidal mechanism for larvae revealed that the inhibition rates of AChE of the three sophoridine derivatives (**SOP-2g**, **SOP-2q**, and **SOP-2r**) at the highest concentration of 250 ppm were 63.16%, 46.67%, and 35.11%, respectively, and **SOP-2g** retained a 44.72% inhibition rate even at the lowest concentration of 50 ppm. Morphological observation of the dead larvae treated with **SOP-2q** and **SOP-2r** showed that the internal structure of the caudal gills was wrinkled and the intestinal cavity and segments obviously collapsed. The abdomen was also thinner and longer with the contents squeezed out due to the deformation and extrusion of the intestinal tract. In summary, this work offered a substantive foundation for identifying efficient larvicidal agents from natural products. We provided a basis for the onward development of sophoridine acting as a natural alternative to control mosquito populations.

## Data Availability

The dataset utilized in this study is available upon request.

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
