# Peer review of "Novel Sophoridine Derivatives as Potential Larvicidal Agents against Aedes albopictus: Synthesis, Biological Evaluation, Acetylcholinesterase Inhibition, and Morphological Study"

_insects, 2023, doi:10.3390/insects14040399_

Round 1

Reviewer 1 Report

This paper presents a comprehensive study on the synthesis and evaluation of sophoridine derivatives for their larvicidal and adulticidal activities against Aedes albopictus mosquitoes. The authors thoroughly examine the insecticidal activities, dose-response relationships, and impacts on the partial life cycle of A. albopictus. They also analyse the structure-activity relationship, investigate the compounds' inhibitory effects on acetylcholinesterase (AChE), and observe morphological changes in dead larvae.

The introduction is well-structured, providing a clear background and objectives for the study. The authors should consider offering more context on AChE's importance in mosquitoes and its role in the insecticidal mechanism. Additionally, a clearer statement of the study's hypothesis and the reasons for choosing sophoridine over other phytochemicals is recommended. Lines 77-79 require clarification regarding previous research on sophoridine and its anti-mosquito effects. Some minor grammatical errors and inconsistencies in punctuation should also be addressed.

The materials and methods section are mostly well-written and detailed. However, further information on the mosquito strain, its resistance status, origin, maintenance, and blood feeding regime would be beneficial. Also, the reason for the photoperiod discrepancy between testing and maintenance should be explained. The positive control concentration for deltamethrin across bioassays should be included, and a clear description of the dose-response analysis methodology should be provided.

The results section is well-organized and presents the data clearly. For dose-response curves, the authors are encouraged to use probit or logit regression. In Figure 1, the nature of the error bars should be explicitly stated. Although median lethal concentrations and confidence intervals are included in Table 3, it would be helpful to present the confidence interval values graphically as well. For probit and logit analysis, the inclusion of goodness-of-fit measures, such as the Chi-squared (χ²) value, is expected. The provided R² values should be discussed in the context of the total variation of the data for the dose-response analysis.

Regarding morphological observations, increasing the sample size from three per treatment group would improve the robustness of the findings. Additionally, a comment on the prevalence of observed phenotypes across bioassayed larvae would be informative.

Overall, the paper is well-executed, with only minor revisions needed for clarity and completeness. Addressing these concerns will strengthen the study and contribute valuable insights to the field of insecticidal research against Aedes albopictus mosquitoes.

Reviewer 2 Report

This manuscript studied the synthesis of sophoridine derivatives and their insecticidal activity against Aedes albopictus. This study aims to explore phytochemicals from plants with larvicidal properties as alternatives for traditional chemical insecticides.  They focused on the derivatives of sophoridine, a natural quinolizidine alkaloid identified from traditional medicinal herbs with biological activities, and finally find three derivatives with oxime ester group SOP-2g, SOP-2q, SOP-2r exhibit larvicidal activity.  Though the results sound scientific significance, two critical issues and some minor questions have to be clarified.  

Critical issues:

1.        The manuscript concluded that unsubstituted at N1 position of the target derivatives maintained the larvicidal activity.  However, the two derivatives SOP-3 and SOP-3f is N-oxides that oxidation of nitrogen-atom at N1 position and exhibit high larvicidal activity (85.333% and 100% respectively).  Please clarify?

Moreover, Line 230-233: After the pre-experiment screening, derivatives with a lethality of at least 50% were chosen to participate in the semi-inhibitory concentration test.  However, the lethality of SOP-3 and SOP-3f were 85.333% and 100% respectively and not included in the semi-inhibitory concentration test.

 2.        Though some sophoridine and its derivatives exhibited potential larvicidal activity against Aedes albopictus, it had low activity against adult mosquitoes.  The authors explain the discrepancy is due to (1) the insecticidal activity of sophoridine derivatives was performed using topical methods against female mosquito and (2) the hydrophilicity of sophoridine compounds.  There are some problems with this explanation. First. for the topical method, the dosage unit used (1000 ppm) is not appropriated. It should be converted to ng or µg/female.  Second, the bioassay conditions for larvae and adult mosquito are different.  For example, whole larva were immersed in 1000 ppm solutions, however, the female adult only topically accepted 0.25 µl 1000 ppm solution.

Minor questions:

1.      Line 383-384: However, the toxicity of the compounds is a well-known way of premedication, and studying them could affect the results of insecticidal activity. I don’t understand what’s mean this sentence related to the insecticidal activity.   Moreover, Line 385-386: For these series of SOP derivatives, their high polarity and hydrophilicity could lead to activity bias toward the larvae rather than the adult.  Since they are hydrophilicity, there should be no need to use acetone as a solvent (Line 217: each derivative was dissolved in acetone).

2.        The experimental sample size is too small, for example, there is only 5 larvae was used for larvicidal activity assay. 

3.        Line 223-224: the sentence ‘Deltamethrin and acetone replaced the derivative as negative and positive control groups, respectively’ should be revised to ‘Deltamethrin and acetone replaced the derivative as positive and negative control groups, respectively’

4.        Line 425: whereas the eclosion rate of the negative control group reached 22% at this time point. It should be 33%, not 22%.

5.        Line 304: what is MPBS buffer. Line 306: the tube was centrifuged at 17xg.  Is it correct?

6.  There is minor mistake in the reference format. For example, reference 4 & 46.

Reviewer 3 Report

This work is devoted to novel sophoridine derivatives as potential larvicidal agents against Aedes albopictus. The synthesis, biological evaluation, acetylcholinesterase inhibition, and morphological study are presented. The work is of interest because this study implies that sophoridine and its novel derivatives could be used to control the population of mosquito larva, which may also be effective alkaloids to reduce the mosquito population density. Taking into account the mentioned below notes, I think that the article looks like a short communication and may be published after major revision.

Notes:

1. The sophoridine derivatives yields should be presented in the Scheme 1 for clarity.

2. The quality of the Figure 2 should be increased. In addition, the captions should be increased for clarity.

3. Why did the insecticidal activity against the larvae of A. albopictus significantly increase when the oxime ester fragment was introduced at the C14 position of sophoridine? What is the role of the oxime ester fragment? It should be discussed more detailed in the text.

4. It would be better to compare the anti-mosquito activity of the known synthetic insecticides with prepared by authors sophoridine derivatives. And the results of this analysis should be presented in Conclusion.

Round 2

Reviewer 2 Report

Though the manuscript has made some revision, there are still some problems for section 2.8.2. Insecticidal tests for female mosquitoes of Ae. Albopictus.  The manuscript described “First, derivatives with lethality greater than 50% for female adult mosquitoes of Ae. albopictus at 1000 ppm were selected to determine the LC50 concentration, which was regarded as the final concentration in this experiment”.  The problems are:

1.      According to Table 2, no derivatives with lethality greater than 50% for female adult mosquitoes of Ae. albopictus at 1000 ppm (should be 0.25 µg/female ). Therefore, no need to perform Insecticidal tests for female mosquitoes.

2. No LC50 for female mosquito of Ae. albopictus was provided in the current manuscript.

Reviewer 3 Report

 Accept in present form

Author Response

Dear Reviewer,

Many thanks again for your comments concerning our manuscript entitled “Novel sophoridine derivatives as potential larvicidal agents against Aedes albopictus: Synthesis, biological evaluation, acetylcholinesterase inhibition, and morphological study”. We have checked our manuscript carefully again and made some revision on English language and style. Revised parts are marked in red in the 'Revised Manuscript with Track Changes' file.

Round 3

Reviewer 2 Report

Since LD50 experiment was not conducted, I suggest to revise Line 254-260 as following:

In order to determine the insecticidal activity toward female mosquitoes of Ae. albopictus, a 0.25 μL (equivalent to 0.25 μg/female) of the 1000 ppm aforementioned derivative solution was topically dropped onto the dorsum of adult mosquitoes with a handheld pipetting device [36, 37]. The derivatives were dissolved in acetone, a highly volatile substance that stayed on the epidermis of adult mosquitoes for a short period of time.

Author Response

Dear Reviewer,

Many thanks for your comments concerning our manuscript entitled “Novel sophoridine derivatives as potential larvicidal agents against Aedes albopictus: Synthesis, biological evaluation, acetylcholinesterase inhibition, and morphological study”. We have looked into your useful and constructive suggestion carefully and made some revision accordingly. Revised parts are marked in red in the 'Revised Manuscript with Track Changes' file. Response to address your concerns are outlined below:

  1. Since LD50 experiment was not conducted, I suggest to revise Line 254-260 as following: In order to determine the insecticidal activity toward female mosquitoes of Ae. albopictus, a 0.25 μL (equivalent to 0.25 μg/female) of the 1000 ppm aforementioned derivative solution was topically dropped onto the dorsum of adult mosquitoes with a handheld pipetting device [36, 37]. The derivatives were dissolved in acetone, a highly volatile substance that stayed on the epidermis of adult mosquitoes for a short period of time.

Response: Thank you for your suggestion. We have revised the manuscript according to your suggestion as showed in the lines 254-258 in the “Revised Manuscript with Track Changes” file.